# Sequencing during Times of Change: Evaluating SARS-CoV-2 Clinical Samples during the Transition from the Delta to Omicron Wave

**DOI:** 10.3390/v14071408

**Published:** 2022-06-28

**Authors:** Shuchen Feng, Mudassir S. Ali, Monika Evdokimova, Gail E. Reid, Nina M. Clark, Susan L. Uprichard, Susan C. Baker

**Affiliations:** 1Department of Microbiology and Immunology, Stritch School of Medicine, Loyola University Chicago, Chicago, IL 60153, USA; sfeng3@luc.edu (S.F.); mali20@luc.edu (M.S.A.); mevdokimova@luc.edu (M.E.); suprichard@luc.edu (S.L.U.); 2Department of Medicine, Stritch School of Medicine, Loyola University Chicago, Chicago, IL 60153, USA; greid@lumc.edu (G.E.R.); nmclark@lumc.edu (N.M.C.); 3Infectious Disease and Immunology Research Institute, Stritch School of Medicine, Loyola University Chicago, Chicago, IL 60153, USA

**Keywords:** SARS-CoV-2, variants of concern, Delta variant, Omicron variant, Artic primer scheme, vulnerable population

## Abstract

The pandemic of SARS-CoV-2 is characterized by the emergence of new variants of concern (VOCs) that supplant previous waves of infection. Here, we describe our investigation of the lineages and host-specific mutations identified in a particularly vulnerable population of predominantly older and immunosuppressed SARS-CoV-2-infected patients seen at our medical center in Chicago during the transition from the Delta to Omicron wave. We compare two primer schemes, ArticV4.1 and VarSkip2, used for short read amplicon sequencing, and describe our strategy for bioinformatics analysis that facilitates identifying lineage-associated mutations and host-specific mutations that arise during infection. This study illustrates the ongoing evolution of SARS-CoV-2 VOCs in our community and documents novel constellations of mutations that arise in individual patients. The ongoing evaluation of the evolution of SARS-CoV-2 during this pandemic is important for informing our public health strategies.

## 1. Introduction

Severe acute respiratory syndrome coronavirus 2 (SARS-CoV-2), the agent causing the Coronavirus disease 2019 (COVID-19) pandemic, has been spreading globally over the past two years and has been evolving to generate multiple variants, such as Alpha, Beta, Gamma, Delta and most recently, the Omicron variant [1]. Since the beginning of the pandemic, the genomic surveillance of SARS-CoV-2 variants of concern (VOCs) has been a continuous effort worldwide [2,3]. Such surveillance effort relies on academic laboratories and public health research laboratories performing high-throughput sequencing of isolated viral RNA and bioinformatics analysis.

Sequencing data collected from a random sampling of SARS-CoV-2-infected subjects in the general population is useful for identifying the evolution and spread of variants across geographical regions over time [2,3]. In addition, analysis of SARS-CoV-2 sequences from hospitalized and/or immunosuppressed patients is of particular interest for monitoring virus evolution during this pandemic. Studies have suggested that immunosuppressed patients, such as solid organ transplant recipients, may not have robust enough antibody responses post-vaccination, making them vulnerable to breakthrough infections and severe COVID-19 disease outcomes [4,5,6]. Ongoing virus replication in these hosts may drive the virus’s genetic evolution, for example, via the emergence of therapy escape mutants or highly transmissible VOCs [7,8,9,10]. One obstacle to such studies is that information about the virus-infected host is usually very limited in publicly available SARS-CoV-2 sequencing data, such as those in the Global Initiative on Sharing Avian Influenza Data (GISAID). As of April 2022, only about 4% of the SARS-CoV-2 sequences in GISAID were associated with “patient status”, which refers to the age and gender of the patients, and additional information about hospitalization and outcome. Since 30 November 2021, the Omicron variant has been classified as a VOC co-present with the Delta variant in the U.S. [1,11]. The Omicron variant and its emerging sublineages, (e.g., BA.1 and BA.2) are characterized by an increased number of mutations in the spike gene (>30 amino acid substitutions) that allow for evasion of vaccine-induced immunity, along with specific changes outside of the spike gene, (e.g., non-structural and nucleocapsid genes) that may play important roles in facilitating viral replication [11,12,13]. Omicron VOCs have been associated with reduced pathogenesis in animal model systems [14,15,16], but higher rates of transmission compared to the Delta variant [17,18,19]. Evaluating SARS-CoV-2 sequencing data along with patient status information may provide insights into drivers of viral evolution in this particularly vulnerable population.

Current analysis of SARS-CoV-2 sequencing data often aims to generate consensus sequences and identify mutations associated with specific SARS-CoV-2 lineages [20,21]. However, patient-specific virus evolution and potential RNA recombination events that may occur during mixed infections may be missed by restricting the analysis to consensus sequences. For example, previous studies have revealed the presence of intra-host single nucleotide variants (iSNVs) of SARS-CoV-2 in infected patients [20,22,23,24]. These iSNVs are often present as host-specific mutations and can occur at low levels (as low as 1% frequency) in the sequencing data and may be missed during the analysis of consensus sequences. Thus, for SARS-CoV-2 clinical samples, a combined analysis of both consensus sequences and sub-consensus level sequencing data (mutations) could be useful for revealing host-specific genetic features that affect viral evolution, pathogenicity, and infectiousness [21,23].

Here, we focused on SARS-CoV-2 clinical samples obtained during the transition from the Delta to Omicron wave (late 2021 to early 2022). These samples were obtained from a majority of immunosuppressed patients with severe disease requiring admission to the hospital or the intensive care unit (ICU) at our medical center in the Chicago, USA area. We reasoned that these patients were likely to have prolonged virus replication, thus having the potential to generate novel mutations. We isolated viral RNA and performed short read amplicon sequencing. We used two different primer schemes, the ArticV4.1 and VarSkip2 primer schemes, to compare their performances (breadth of coverage) on sequencing potential Delta and Omicron samples, as these two primer schemes were both specifically modified from their earlier versions to incorporate in the mutations of the Omicron variant. We then performed sequence analysis at both consensus and sub-consensus levels. Additionally, we examined the genetic features and identified mutations unique to each patient sample. Overall, our study describes successful amplicon sequencing results for recently emerged variants and outlines a strategy of multistep bioinformatics analysis of SARS-CoV-2 variants that facilitates identifying lineages and novel genetic features that may contribute to future emerging variants of concern.

## 2. Materials and Methods

### 2.1. Sample Collection and RNA Extraction

Nasopharyngeal swab specimens collected from patients at our medical center were provided for this study using a protocol approved by the Loyola University Health Sciences Campus Institutional Review Board (IRB# 214365). After collection, samples were aliquoted and stored at −80 °C. Before extraction, samples were heat-inactivated at 55 °C for 30 min. RNA was extracted from 200 μL of each sample using the MagMAX Pathogen RNA/DNA Kit (Applied Biosystems, Foster City, CA, USA) on a KingFisher Flex automated system (Thermo Fisher Scientific, Inc. Waltham, MA, USA). RNA was eluted in 90 μL of RNase-free water.

### 2.2. Evaluating SARS-CoV-2 Viral Load in Patient Samples Using RT-qPCR

Quantification of SARS-CoV-2 RNA was performed using reverse transcription-quantitative PCR (RT-qPCR) and the N1 assay according to the instruction of CDC 2019-nCoV Real-Time RT-PCR Diagnostic Panel [25] with one revision in the amplification program (see below). Briefly, a total reaction volume of 20 μL was used, including 5 μL of the TaqPath^TM^ 1-Step RT-qPCR Master Mix, CG (Thermo Fisher Scientific, Inc. Waltham, MA, USA), 8.5 μL DNase/RNase-free water, 5 μL of RNA template, primers at a final concentration of 500 nM and probe at a final concentration of 125 nM (2019-nCoV RUO Kit, Integrated DNA Technologies, Inc, Coralville, IA, USA). The amplification program started at 25 °C for 2 min, followed by 50 °C for 15 min, 95 °C for 2 min, then 45 cycles of 95 °C for 15 s and 55 °C for 30 s. A standard curve was established by running six dilutions of SARS-CoV-2 Plasmid Controls (Integrated DNA Technologies, Inc, Coralville, IA, USA) in triplicate from 10^6^ copies to 10 copies per reaction. The N1 assay had a slope of −3.373, a y-interception of 42.216, and an efficiency of 97.9%. A human RNase P (RP) assay (2019-nCoV RUO Kit) was also performed for each sample with the same program as a quality control for clinical samples. No template controls (negative control) and known positive controls were also included in each run. All samples with cycle threshold (Ct) values of less than 30 were subjected to amplification and Illumina amplicon sequencing, along with two samples that have Ct values of 31 (sample P18) and 33 (sample P15), respectively, to evaluate the performance of sequencing on samples with low levels of viral RNA.

### 2.3. Sequencing Library Generation and Sequencing

Amplicon library generation was performed using the NEBNext ARTIC SARS-CoV-2 FS Library Prep Kit for Illumina according to the manufacturer’s instructions (E7658L, New England Biolabs, Ipswich, MA, USA). Briefly, RNA samples were first reverse transcribed to cDNA using the LunaScript™ RT SuperMix (included in the kit). Amplification of cDNA was performed using Q5 Hot Start High-Fidelity master mix (included in the kit) with the ArticV4.1 (catalog #10011442, Integrated DNA Technologies, Inc, Coralville, IA, USA) or VarSkip2 (included in the kit) primer schemes. The ArticV4.1 library preparation was performed as instructed in the standard version of protocol. The VarSkip2 scheme library preparation was performed according to the express protocol provided by the manufacturer. Sequencing was performed using an Illumina Miseq with the V2 reagent kit in 2 × 150 base pair (bp) read length. A total of 37 RNA samples from 34 patients were sequenced using the ArticV4.1 primer scheme in a single run, with a subset of eight samples using VarSkip2 in the same run (P1, P2, P15, P17, P18, P24, P30, and P31). All 37 RNA samples using ArticV4.1 and the eight using VarSkip2 reached near-full-genome coverage.

### 2.4. Bioinformatics Analysis Flow

Raw reads were assessed for quality using FastQC (https://www.bioinformatics.babraham.ac.uk/projects/fastqc/, accessed on 22 March 2022), followed by adaptor trimming in cutadapt [26] and quality trimming in BBduk (http://jgi.doe.gov/data-and-tools/bb-tools/, accessed on 22 March 2022). Bwa-mem [27] was used for paired-end reads mapping to the Wuhan-Hu-1 reference genome (NCBI RefSeq accession NC_045512.2). Primer trimming was performed with iVar [28] using bed files specific to Artic V4.1 or Varskip2, respectively. Trimmed bam files were then realigned, deduplicated for final coverage evaluation, and called for variants using “ivar variants −q 20 −t 0.3 −m 30”. Consensus sequences were generated using “ivar consensus −q 20 −t 0.3” and analyzed through Nextclade for private mutation information; all private mutations identified from consensus sequences of each sample were also compared to mutation calling results and/or the corresponding BAM file to ensure the accuracy of consensus calling. A phylogeny tree was generated using the Nextstrain [29] SARS-CoV-2-specific procedures and visualized in auspice (https://auspice.us, accessed on 6 April 2022). The iVar output files were converted to variant call format (VCF) files and further processed in bcftools (v1.12) [30] and SNPeff (v5.0) [31] for formatting and annotation, respectively.

### 2.5. Read Frequency Cut off

We checked the total mutation numbers at read frequencies of 0.01, 0.03, 0.1, 0.3, 0.5, 0.7, 0.8, 0.95, and 1 for the eight samples sequenced using both primer schemes (Appendix A). Briefly, we called mutations for each of the eight samples at the listed read frequencies using iVar (command “ivar variants”), and then summarize the total mutation numbers in the eight samples at each read frequency. We determined the reliable read frequency cut-off was at 0.3 for our ArticV4.1 samples, which had a higher depth than VarSkip2, based on the following observations: (1) both ArticV4.1 and VarSkip2 primer schemes had relatively stable total numbers of mutations called from 0.1–0.7 read frequency cut off, although ArticV4.1 with higher depth showed slightly higher total numbers than Varskip2 (Appendix A); (2) at 0.3 level, variant-specific mutations could be found in all samples belonging to this variant, such as the Omicron BA.1 and BA.1.1 specific mutation G8393A.

### 2.6. Evaluating a Sample Specific ORF7a Region Deletion Using RT-PCR and Sanger Sequencing

Amplicon sequencing revealed a putative deletion in the ORF7a region in sample P2. To further evaluate the ORF7a region, we subjected viral RNA isolated from samples P2 (12.4 ng/µL) and P10 (8.5 ng/µL) to RT-PCR using 11 µL of RNA and random hexamers to generate cDNA (RevertAid First Strand cDNA Synthesis Kit, K1621, ThermoFisher Scientific, Inc. Waltham, MA, USA). Sample P2 was the Delta sample indicated to have low-to-no coverage ORF7a region in both ArticV4.1 and VarSkip2 amplicon sequencing results. Sample P10 was another Delta sample indicated to be complete in the ORF7a region from the amplicon sequencing result, and therefore served as a positive control in this experiment. A PCR reaction was performed using Q5^®^ High-Fidelity DNA polymerase (M0494S, NEB, Ipswich, MA, USA). The 50 µL reaction was performed using 2 µL of cDNA, 2.5 µL of 10 µM primer PCR pF, 2.5 µL of 10 µM primer PCR pR, and 25 µL of 2X Master Mix under the following thermocycling conditions: 98 °C for 30 s, 35 cycles of 98 °C for 10 s, 65 °C for 30 s, and 72 °C for 40 s, and a final extension at 72 °C for 5 min. For gel electrophoresis, 2–4 µL of PCR product was run on a 1.5% agarose gel stained with ethidium bromide at 60V for 1.5 h. For visualization, the FluorChem system was used (ProteinSimple, San Jose, CA, USA). The remaining PCR products were purified using the Wizard^®^ SV Gel and PCR Clean-Up System (Promega, Madison, WI, USA) and subjected to Sanger sequencing (ACGT, Inc., Wheeling, IL, USA) using the sequencing primers listed in Table 1.

### 2.7. Mutation and Statistical Analysis

All mutation and statistical analyses were performed in R v4.0.5 [32]. Permutation tests to associate mutations and patient groups were performed in R package indicspecies [33].

## 3. Results

### 3.1. Evaluating SARS-CoV-2 Genomic RNA in Nasal-Pharyngeal Swabs from Patients with Breakthrough Infections

We previously reported on the analysis of SARS-CoV-2 clinical samples associated with breakthrough infections with SARS-CoV-2 variants of concern in March–April 2021 [34]. We found that: (1) some vaccinated individuals were susceptible to infection with new variants even after receiving mRNA vaccines; (2) immunosuppressed patients were more likely to be hospitalized after breakthrough infection; and (3) diverse variants were associated with these breakthrough infections. Our report agreed with other reports and highlighted the importance of monitoring viruses associated with breakthrough infections, particularly in patients with severe disease [35,36,37]. Here, we continue our analysis of SARS-CoV-2 RNA isolated from a majority of older (>age 50, 79%) and immunosuppressed patients (79%) who experienced breakthrough infections as summarized in Table 2. Of the 34 patients in this study, 28 were hospitalized for COVID (82%), with 14 patients (41%) admitted to the ICU. We analyzed a second sample obtained from three patients (P14, P16, and P29) on day 10–17 after the initial sample, (i.e., a total of 37 RNA samples), to investigate the potential evolution of the virus during prolonged replication. For each sample, viral RNA was isolated and subjected to RT-qPCR analysis to determine the viral load (indicated by Ct values). The Ct values ranged from 17.8 to 33.1. We found no statistically significant difference in the viral load in immunosuppressed versus normal patients (Welch’s *t*-test, *p*-value = 0. 344) (Appendix A). We report that we generated amplicon libraries from all samples with Ct values less than 33 (N1 assay). These amplicon libraries were subjected to sequencing using Illumina Miseq.

### 3.2. Bioinformatics Analysis of the SARS-CoV-2 Genomes Identifies Lineages and Sublineages

Most of our samples were obtained during the period from November 2021 to February 2022, during the transition from Delta VOC to Omicron VOC in the Chicago area. We were concerned about the performance of primer amplification systems that would maximize coverage of both Delta and Omicron variants. To address this concern, we evaluated the performance of both the ArticV4.1 and VarSkip2 primers using a subset of eight samples to allow for direct comparison, with the remaining 29 samples amplified using ArticV4.1. All 37 RNA samples were successfully amplified and sequenced using the ArticV4.1 primer scheme and achieved near-full-length genome breadth of coverage with an average of 99.82% ± 0.42% (mean ± SD) at an average depth at 2024X ± 371X (mean ± SD) (Figure 1). No significant correlation was observed between viral load, (i.e., Ct value, ranged from 17.8 to 33.1) and breadth of coverage (Spearman’s rho = −0.15, *p*-value = 0.312). The VarSkip2 primer scheme had an average breadth of coverage of 99.78% ± 0.28% at a relatively lower average depth of 525X ± 116X, which was likely because of the express VarSkip2 library preparation method with less template input as instructed by the manufacturer. Overall, both the ArticV4.1 and VarSkip2 primer schemes showed near-full-genome coverage, despite the difference in sequencing depth and therefore we used both data sets for lineage identification.

To identify the SARS-CoV-2 lineages in the 37 RNA samples, we generated consensus sequences from each sample and classified lineages based on results from Pangolin (v3.1.2) (https://pangolin.cog-uk.io, accessed on 22 March 2022) and Nextclade (https://clades.nextstrain.org, accessed on 6 April 2022) (Appendix A). Three variants were characterized; one sample from 26 May 2021, was identified as Alpha (B.1.1.7 lineage), 10 samples collected from 5 August to 31 December 2021, were identified as Delta variants (AY sublineages), and 26 samples from 15 December 2021 to 1 February 2022, were identified as Omicron variants (BA.1, BA.1.15 and BA.1.1 sublineages) (Figure 2A, Appendix A). As expected, the Omicron samples had more mutations than the Delta samples, with an average of 57.4 substitution mutations compared to Delta which had an average of 44.8 substitutions (Figure 2A). For the patients sampled twice (10–17 days apart, Appendix A), we determined that the virus had identical lineages (P14, P16, and P29). For Omicron samples, 10/24 patients (42%) had severe outcomes (ICU), 9/24 (38%) were hospitalized but not in ICU, and five were not hospitalized (21%). For Delta samples, 3/9 were in ICU (33%), 5/9 were hospitalized but not in ICU (56%), and one was not hospitalized (11%). No statistical difference was observed between the Omicron and Delta patient status in our sample set (Student t-test, *p*-value = 0.5). We observed a clear lineage transition from Delta to Omicron VOC in December 2021 within our set of 34 patients, which corresponds to the transition from Delta to Omicron from late 2021 to early 2022 in the state of Illinois, U.S. (Figure 2B). In addition, for the subset of eight samples that were sequenced using VarSkip2, we found the same lineages as with ArticV4.1: six Omicron (P15, P17, P18, P24, P30, and P31), one Delta (P2) and one Alpha (P1). Therefore, we conclude that for lineage identification purposes, both primer schemes are useful, even with a lower read depth of ~500X for the VarSkip2 sequenced samples. Our lineage analysis highlights the rapid transition from the Delta to Omicron VOC in the Chicago area and suggests a potential for co-infection of Delta and Omicron during the time when both variants were circulating [38].

### 3.3. Assessment of Sequencing Outcomes Reveals a Sample-Specific Deletion in a Delta Variant

We then evaluated our sequencing outcomes for the patterns of coverage in the Delta and Omicron variants samples, respectively. The ArticV4.1 primer scheme showed very similar patterns for both Delta and Omicron variants, with greater than 30X coverage for the entire genome (red line in Figure 3). We did note several regions with slightly reduced coverage (amplicons #5, #8, #21, #23, #31, and #74), which were previously noted for reduced performance by reports of the Artic primer scheme [39,40]. In our outcomes using ArticV4.1, amplicon 74 had the lowest coverage at average depths of 168X ± 131X and 85X ± 67X for Omicron and Delta samples, respectively. The Varskip2 sequencing results showed patterns of low coverage regions at amplicon 9 and from amplicon 44 to amplicon 49, but still greater than 30X coverage for each genome (Appendix A).

Since we noticed the presence of low coverage regions, we further evaluated the sequencing breadth of coverage for each sample to make sure no sample-specific dropout or deletion was present. We observed in one Delta sample (P2) a unique 227 bp low-to-no coverage region corresponding to ORF7a in both the ArticV4.1 and VarSkip2 primer schemes’ results, which did not correspond with any single amplicon region (Figure 4A). To determine if this region in P2 was a sample-specific deletion, we designed RT-PCR primers to amplify the region of interest in P2 as well as in another full-length Delta sample, P10, which served as a positive control (Figure 4B). Gel electrophoresis of the PCR products was consistent with the expected amplicon sizes of 638 bp and 865 bp for P2 and P10, respectively (Figure 4C). Sanger sequencing of the RT-PCR products revealed that P2 had a 227bp deletion along with two amino acid substitutions in the N-terminal region of ORF7a. The Sanger sequencing was identical to the ArticV4.1 and Varskip2 amplicon sequencing of this region. This 227-nucleotide deletion in ORF7a results in a frameshift mutation that truncates ORF7a to 42 amino acids (Figure 4D). In contrast, both the reference SARS-CoV-2 genome and the control P10 Delta sample showed an intact ORF7a of 121 amino acids. Deletions in the ORF7-ORF8 region have been previously identified from patient samples [41,42], but seem to arise sporadically and are not maintained in the population (see discussion).

### 3.4. Analysis of Unique Mutations in Omicron and Delta Samples on Consensus Sequence Level

We then evaluated the host-specific mutations within our Delta and Omicron samples. We first analyzed the consensus sequences through Nextclade to identify the private mutations for each patient (Figure 5, Appendix A). Private mutations are mutations that are not observed in the nearest neighbor on the Nextclade’s reference tree and are therefore very likely to be specific to individual samples. Among 33 patients with Delta and Omicron variants, 31 were identified to have private mutations, including nine Delta patients (100% of total Delta patients) and 22 Omicron patients (85% of total Omicron samples), indicating the prevalence of SARS-CoV-2 genome variability in clinical samples (Figure 5A). More private mutations were observed in Delta than Omicron variants in our dataset, with an average of ten private mutations in Delta samples and four private mutations in Omicron samples. Twenty-nine percent of Omicron’s private mutations were “labeled mutations”, which are private mutations that have already been described for a genotype known to be commonly present in a clade. For example, the labeled mutation C21595T was identified in all the Omicron sublineage BA.1.1 samples (*n* = 6); while mutation C28472T was identified in the Omicron sublineage BA.1 sample (*n* = 12). For Delta samples, only 3% of the private mutations were labeled and no association was observed for samples within the same sublineage, such as P5, P6, and P10 which were all lineage AY.3. An unlabeled private mutation of higher prevalence and sublineage association, C11950T, was observed in BA.1 lineage Omicron samples with the co-occurrence of C28472T. This mutation has been identified in Omicron clinical samples from the upper Midwest US (Illinois and Wisconsin) in late 2021 and was designated to be associated with a newer Omicron sublineage BA.1.20 in April 2022 (https://github.com/cov-lineages/pango-designation/issues/375, accessed on 20 April 2022). This updated designation of Omicron sublineage BA.1.20 corresponds to our phylogeny designation that these Omicron samples are placed on different clades than the other two BA.1 samples (P11 and P19) (Figure 2A).

Furthermore, we observed that all private mutation patterns were patient-specific (Figure 5A). We detected identical and repeat private mutation patterns in the same patient sample that was sequenced twice (P14, P16, and P29, Figure 5B), consistent with host-adapted mutations in SARS-CoV-2 variants. Among the three repeatedly sampled patients, two had identical private mutations in both sampling events (P14 and P16). Notably, one patient had an increase from four private mutations to ten private mutations between the first and second sampling events, indicating an accumulation in mutations across the time course of infection in the patient. The viral RNA load remained high in all three patients, as determined by RT-qPCR (Appendix A). These three patients were in the ICU during the second sampling event (10–17 days apart from the first sampling), suggesting that the virus could either remain stable or accumulate mutations over time, consistent with previous reports [7,8].

### 3.5. Analysis of Delta and Omicron Sequences for Evidence of Recombination Events and Association of Mutations with Disease Severity

Coronaviruses have been shown to undergo RNA recombination in cells co-infected with different viral strains [43]. Since Delta and Omicron VOCs co-circulated during the time of our study, we reasoned that there was potential for co-infection [38]. Therefore, we evaluated the SARS-CoV-2 sequences for evidence of RNA recombination. To accomplish this goal, we looked for the presence of Omicron-specific mutations in all Delta samples and vice versa (see mutation list in Appendix A; method as described in [44,45]). In addition, we searched for the presence of Omicron BA.2 lineage-specific mutations in all our Omicron samples (BA.1.15, BA.1.1, and BA.1 sublineages). We found no evidence of RNA recombination events of either Delta/Omicron or Omicron BA.1/BA.2. These results agreed with our private mutation analysis results, which showed completely different private mutation patterns in Delta and Omicron samples (Figure 5A).

In addition to the potential for recombination events, we also evaluated potential associations between identified mutations and the severity of disease in the patient. Studies suggested that missense mutations could be related to the virus’s ongoing adaptation, reduced symptoms, or disease severity, along with changes in other characteristics such as transmission rates [28,29]. Therefore, we examined the impacts of frequencies of silent and missense mutations on four groups of variants and patient outcomes: Omicron-ICU (group A), Omicron-not-ICU (group B), Delta-ICU (group C), and Delta-not-ICU (group D) (Appendix A, Appendix A). On average, the missense mutation’s frequency (71.6% ± 2.6%, mean ± SD) was 2.5 folds of the silent mutation’s frequency (28.3% ± 6.3%) (Student t-test, *p*-value < 2.2 × 10^−16^), which agreed with other studies [46,47] and perhaps indicated the selection process within the hosts. However, no statistically significant difference was observed for either comparison between groups within silent or missense frequencies (Welch’s t-test, see Appendix A), suggesting the individual health outcome is not solely affected by genetic mutations under intra-host selection pressure.

We also wanted to determine if a lineage or a variant-related mutation was associated with disease severity that required admission to the ICU. The patient samples in this study are derived from mostly (82%) hospitalized patients. In agreement with the previous study [34], we found (1) the most significant factor associated with progression to a severe disease requiring ICU admission in our patient group was age (over 60 years old, *p*-value = 0.032) and (2) there was no single sublineage of Delta or Omicron variant associated with admission to the ICU (*p*-value = 0.433). We further examined if there was any correlation between identified mutations and patient health outcomes by dividing the samples into the four groups mentioned above. We performed permutation tests (R package indicspecies [33]) using all mutations in these samples to explore their potential associations with the assigned groups. We report that no single mutation was identified to be specific to, or associated with, severe disease. Our report supports previous findings that known factors such as age over 60, immunosuppression, and other co-morbidities previously identified including diabetes and high body mass have been reported to be more significantly associated with disease severity than any specific SARS-CoV-2 VOC [34,48,49,50,51,52].

## 4. Discussion

Since the early stage of the pandemic, the evolution of SARS-CoV-2 has been shaped by selection pressures from vaccination, antiviral therapies, and other mitigation strategies [7,8,10,53]. The general population in many countries has been protected from severe disease through the distribution of vaccines, while the immunosuppressed population may still be at risk for severe disease despite their vaccination status [9,34,54]. Ongoing virus replication in these patients, as well as virus replication in the general population, may contribute to the evolution of new variants of concern [8,9,10,53].

In this study, we evaluated the genetic features of SARS-CoV-2 variants from patient samples (*n* = 34) obtained during the transition of Delta and Omicron waves. We focused on the analysis of samples from patients who were hospitalized with breakthrough infections (82% of all patients). The majority of these patients were immunosuppressed (79%) due to treatments associated with solid organ transplant, multiple myeloma, systemic lupus erythematosus, or chronic leukemia. This population of patients is highly vulnerable to severe COVID-19 and 41% were admitted to the intensive care unit (ICU). We report low cycle threshold (Ct) values for these patients, suggesting high levels of virus replication (Appendix A). We also found agreement with other findings that older age, (e.g., >60) was a statistically significant factor correlated with severe disease outcomes [34,49,51,52,55]. We identified both Delta and Omicron lineages in hospitalized patients (Figure 2A). This is consistent with the idea that the variants in the general population contribute as the source of infection for vulnerable populations. Additionally, breakthrough infections of Omicron sublineage BA.1 characterized by the two substitutions (C11950T and C28472T, Figure 5A) were also reported in the young and fully vaccinated general population in the Chicago area in late 2021 without hospitalization after infection [56], suggesting that individual immunosuppression may play a bigger role in severe disease outcomes than any specific variant or lineage. Therefore, we conclude that the breakthrough infections in our sample set were associated with the circulating variants of concern at the time of sampling, including the AY sublineages of the Delta variant and the BA.1 and BA.1.1 sublineages of the Omicron variant.

For detailed SARS-CoV-2 genetic feature evaluation, we aimed to identify the viral mutations that could be of clinical and/or evolutionary significance in our sample set. We examined the intra-host mutations detected at relatively high frequencies (≥0.3) rather than including low-frequency mutations (≥0.01), as previous studies have indicated that iSNVs detected at very low levels in individuals are less likely to become associated with new variants of concern [20,23,47]. In agreement with recent studies [9,20,47,57,58], we also found patient-specific private mutation patterns as evidence of iSNVs. The private mutations we observed revealed a remarkable host specificity and intra-host consistency, particularly among three ICU patients that were repeatedly sampled (P14, P16, and P29). The private mutation patterns in these patients were either identical across time or showed mutational accumulation over time (Figure 5B). Additionally, the patient-specific private mutations observed in both Delta and Omicron samples were detected in the S gene and across the whole genome. These results indicate the importance of whole genome sequencing and support a previous finding that mutations in the non-spike regions, such as the N gene, contribute to viral transmission [59]. The majority of the private mutations in our samples were located in ORF1ab, followed by the S gene and N gene, consistent with a report by Li et al. [47]. The normalized private mutation number (per 1kb length per patient) indicates that the Delta samples had the highest number of private mutations in the ORF7a gene (1.5 private mutations), and the Omicron samples had the highest number of private mutations in the N gene (0.6 private mutations), which was consistent with the report from Lythgoe et al. [20]. This distribution pattern of host-specific mutations across the whole genome in different variants and lineages of SARS-CoV-2 documents how iSNVs may contribute to the evolution of SARS-CoV-2 [20,47].

Our analysis also revealed the importance of examining viral evolution in individual patient samples. Both our amplicon sequencing and RT-PCR detected a 227 bp deletion in the ORF7a region in one Delta sample (P2) (Figure 4). Deletions in ORF7a have been identified previously, but are rare [41,42,60,61]. For example, Nemudryi et al. [61] found that out of 180,971 SARS-CoV-2 genomes deposited in GISAID, only 845 contained novel ORF7a gene variants. Although mutations in ORF7a continue to appear sporadically across different SARS-CoV-2 lineages, they likely arise de novo in the patient, as they are rarely transmitted to others and tend to disappear. This is consistent with in vitro work that has shown that mutations in ORF7a may cause viral attenuation, perhaps because of an inability to suppress the interferon response [61,62,63]. This ORF7a region may be required for efficient virus propagation in the host reservoir but not required for replication in humans. In addition, the finding that deletions arise sporadically in the same region of the genome suggests these deletions could be correlated with RNA secondary structure [64,65]. Continuing to monitor deletions in the accessory genes of SARS-CoV-2 may provide insight into the functions of these proteins and the evolutionary adaptations that zoonotic viruses must go through to infect different host species.

In addition to the lineage and variant analysis for SARS-CoV-2 clinical samples, our work also aimed to provide useful technical information for sequencing Delta and Omicron variants. We used two primer schemes, ArticV4.1 and VarSkip2, for sequencing SARS-CoV-2 clinical samples. Both primer schemes are modified for short reads amplicon sequencing for Omicron variants, yet they differ in the amplicon sizes and numbers across the genome. Sequencing using two primer schemes provided us with strong evidence in discerning the 227 bp deletion in the sample P2. This low-to-no coverage region was confirmed to be not an amplicon dropping out issue caused by one of the primer schemes. We found that both ArticV4.1 and VarSkip2 reached near-full-genome coverage (Figure 1) and resulted in identical lineage assignments using both Pangolin and Nextclade. However, for mutation calling analysis, we found that a higher depth was preferred. For example, at the same read frequency of 0.8 (Appendix A), we were able to find all VarSkip2-only mutations in lower read frequency in the ArticV4.1 samples, but the ArticV4.1-only mutations were not present in the VarSkip2 group even when the read frequency was lowered to 0.03, indicating a failure of calling such mutations when depth was low. Therefore, we suggest a depth of ~2000X would be sufficient for mutation calling. We also suggest the rule of least requirement of reading frequency for mutation calling to be reported in studies sequencing SARS-CoV-2.

It is important to acknowledge the limitations of our study. The sample number in our study was relatively small, with a total of 37 samples from 34 patients. Additionally, the patients included in this study were mostly hospitalized and immunosuppressed patients. These limitations led to reduced power in our statistical analysis when evaluating the correlation of patient demographics with the disease outcomes, as well as our analysis of the correlation between mutations and patient health outcomes. Studies that monitor SARS-CoV-2 variants in more broadly representative patient demographics and larger sample numbers are needed to determine if particular genetic features of variants are associated with breakthrough infections that result in hospitalizations of older and/or immunocompromised individuals.

## 5. Conclusions

In this study, we identified the lineages and mutations of SARS-CoV-2 variants associated with breakthrough infections in patients during the Delta and Omicron waves in the Chicago area. We demonstrated the presence of patient-specific SARS-CoV-2 mutations that can be indicators of emerging sublineages in the local communities. These results contribute to our understanding of the ongoing evolution of SARS-CoV-2 that occurs during virus replication in outpatients and hospitalized patients. Surveillance of SARS-CoV-2 is important for understanding the ongoing evolution of this virus and the potential implications for public health.

## Figures and Tables

**Figure 1 viruses-14-01408-f001:**
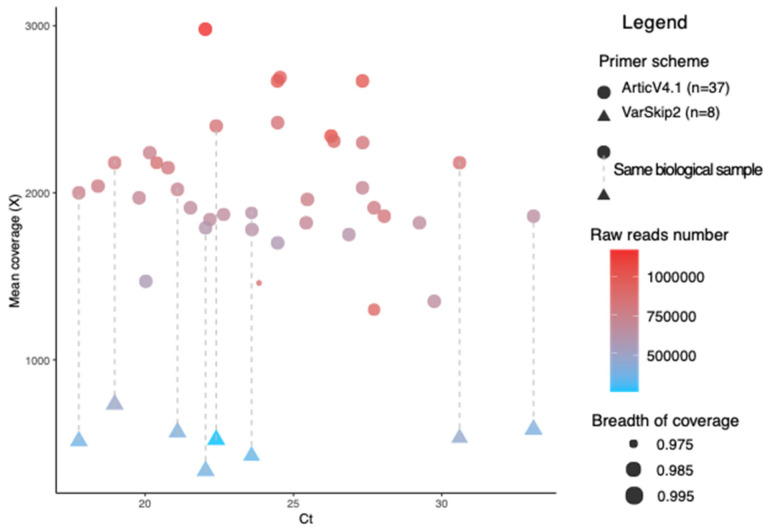
Sequencing outcomes of 37 clinical samples sequenced using the ArticV4.1 primer scheme with a subset of eight using VarSkip2. The *X*-axis shows samples’ Ct values, the *Y*-axis shows the mean coverage (X). Circles and triangles represent samples sequenced using ArticV4.1 and VarSkip2, respectively. The subset samples sequenced using both primer schemes (*n* = 8) are indicated by the connection of gray dashed lines. Color scale shows paired end read numbers in each sample increasing from blue to red. Point sizes represent the breadth of coverage for each sample.

**Figure 2 viruses-14-01408-f002:**
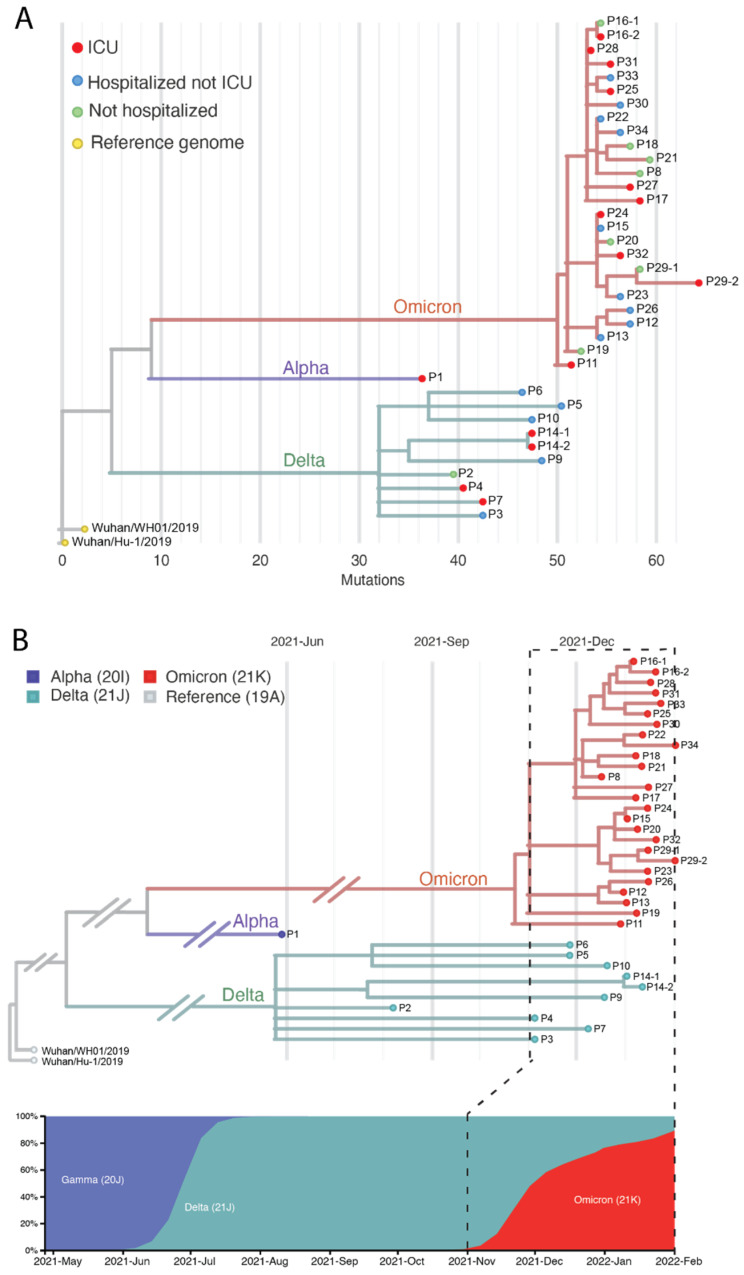
Lineage identification of the 37 clinical samples sequenced in this study. (**A**) Phylogenetic tree of the 37 clinical samples. Variants of Omicron, Alpha, and Delta are indicated in branch colors of red, purple, and green, respectively. Patients of ICU, hospitalized but not ICU, and no admission are indicated in tips in red, blue, and green, respectively. The two reference genomes are indicated in yellow tips. (**B**) Alignment of the identified variants in this study (**top**) and the normalized variant frequencies in the state of Illinois (**bottom**) during the same period. Note the branch lengths in B are not representative of the evolutionary time between two nodes, but rather indicate the variant assignments of the 37 samples and the sampling date.

**Figure 3 viruses-14-01408-f003:**
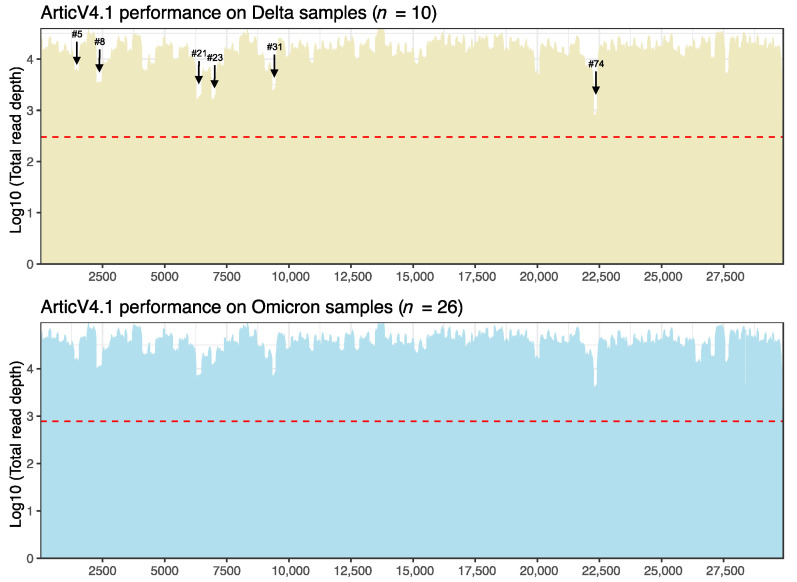
Summarized total breadth of coverage for Delta and Omicron samples sequenced using the ArticV4.1 primer scheme. The *X*-axis represents genome positions. The *Y*-axis represents log10 transformed total read depth for each variant. The red dashed lines indicate the minimum total read depth of n samples for reliable mutation calling, calculated as [the minimum depth required (30X) × number of samples of Delta or Omicron]. Low coverage amplicon regions are marked out in black arrows.

**Figure 4 viruses-14-01408-f004:**
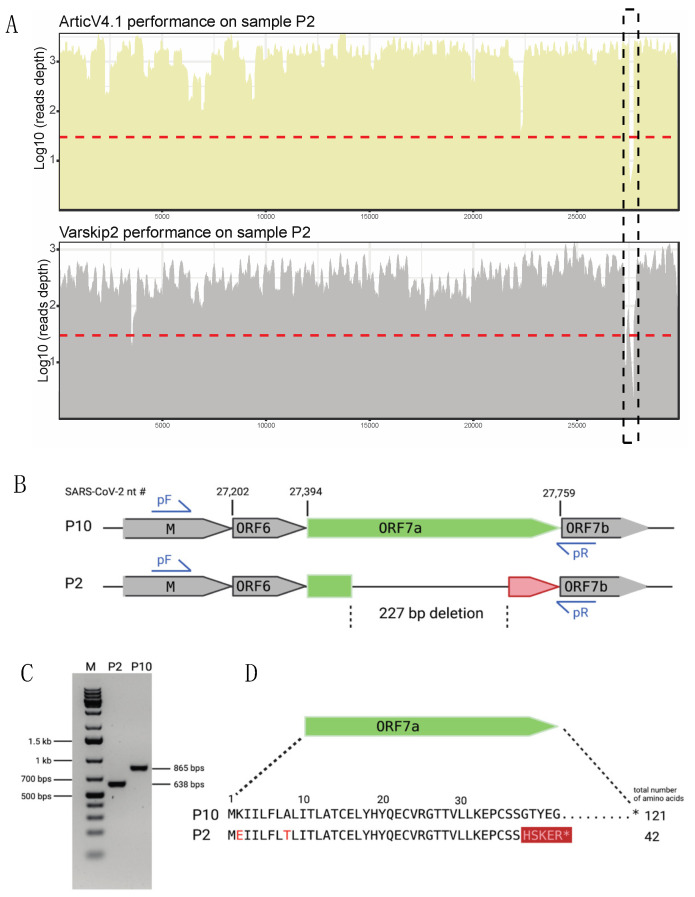
A 227 bp deletion observed in the ORF7a region of one of the Delta samples, P2. (**A**) Amplicon sequencing using both ArticV4.1 and VarSkip2 indicates a deletion in the ORF7a region of sample P2. (**B**) Schematic diagram of the SARS-CoV-2 genome of P2 and a no-deletion Delta sample, P10, highlighting the 227 bp ORF7a deletion observed in P2 (created with BioRender.com). Primers pF and pR were used for RT-PCR to confirm the deletion. The SARS-CoV-2 nt # denotes the nucleotide position in the reference genome (GenBank accession NC_045512.2). (**C**) Gel electrophoresis results of the PCR products showed the expected size difference between P2 and P10. (**D**) Schematic diagram of the Sanger sequencing results showing the predicted ORF7a amino acid sequence of samples P2 and P10 (created with BioRender.com). The resulting frameshift mutation and stop codon in P2 is highlighted in red.

**Figure 5 viruses-14-01408-f005:**
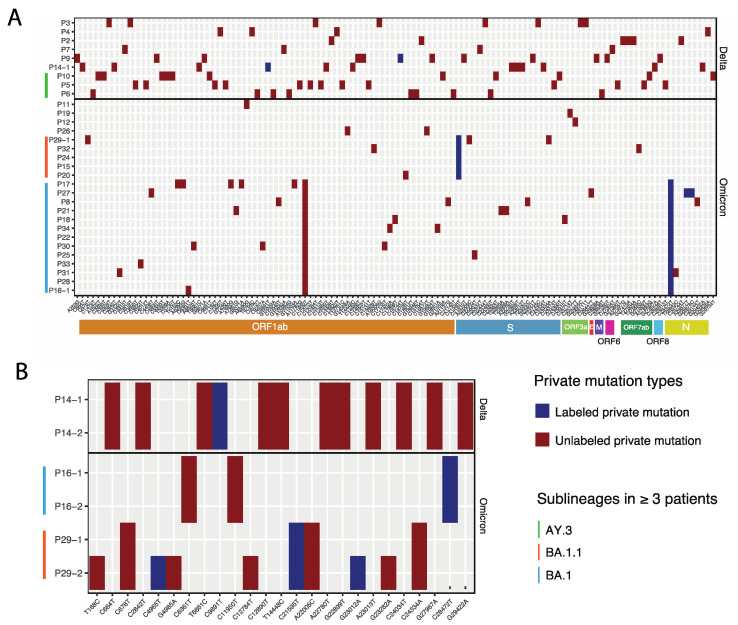
Private mutation patterns observed in Delta and Omicron variants infected patients. (**A**) Private mutations found in applicable Delta and Omicron patients (*n* = 31). *Y*-axis shows the patient samples and *X*-axis shows the exact mutations with their corresponding gene position in the coding regions indicated at the bottom. (**B**) Private mutation patterns in three repeatedly sampled patients (P14, P16, and P29). For both (**A**) and (**B**), labeled and unlabeled private mutations (as identified by Nextclade) are shown in dark blue or dark red colors, respectively. Sublineages of Delta and Omicron identified in multiple patients (n ≥ 3) are indicated on the left of the *Y*-axis in different colors.

**Table 1 viruses-14-01408-t001:** RT-PCR and Sanger sequencing primers for the identified ORF7a deletion.

Primers	Primer Sequence (5′ to 3′)
PCR pF	AACACAGACCATTCCAGTAGC
PCR pR	GACACGGGTCATCAACTACAT
Sequencing Primer pF1	CACTGATAACACTCGCTACTTG
Sequencing Primer pF2	GCTTTGCTTGTACAGTAAGTGAC
Sequencing Primer pR3	TGCAGCTACAGTTGTGATGAT
Sequencing Primer pR4	TGCAGTTCAAGTGAGAACCA

**Table 2 viruses-14-01408-t002:** Summary of patient demographics (*n* = 34).

		*N* or Range	% or Median
Sex	Male	20	59%
	Female	14	41%
Age Range (Median)	Male	30–79	55
	Female	30–67	62.5
Vaccination Status	Vaccinated (≥1 dose)	33	97%
	Unvaccinated	1	3%
Immunosuppression Status	Immunosuppressed *	27	79%
	Normal	7	21%
Hospitalization Status **	Hospitalized	28	82%
	Not hospitalized	6	18%

* Immunosuppression due to multiple myeloma, solid organ transplant, systemic lupus erythematosus, or chronic lymphoid leukemia. ** The hospitalized group includes 14 patients in ICU. Patients who progressed from outpatient to hospitalization are included in “Hospitalized”. Patients admitted for non-COVID reasons (asymptomatic, P18, P19, P21) were treated as “Not hospitalized” for COVID.

## Data Availability

The complete genome sequences of the SARS-CoV-2 variants sequenced in this study were deposited to the GISAID database under accession numbers EPI_ISL_12690971 to EPI_ISL_12691015.

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
