# Peer review of "Sequencing during Times of Change: Evaluating SARS-CoV-2 Clinical Samples during the Transition from the Delta to Omicron Wave"

_viruses, 2022, doi:10.3390/v14071408_

Round 1

Reviewer 1 Report

The study shows the mutational analysis of SARS-CoV-2 in 37 clinical samples from immunocompromised patients. The two primal schemes were also compared for mutational analysis. Overall, the study is well-conducted and worth reporting.

I have a few comments for the authors:

1)      Line 200) states “All sample concentrations were sufficient for generating amplicon libraries for sequencing studies”. What do you mean by sample concentration? Volume? Or concentration of RNA?

2)      In the introduction part, I would suggest adding more about the two primer schemes. Why were they chosen? What is the difference between the 2 to make it clear for the reader? What are the advantages of studying or comparing them for future studies?

3)      One of the biggest limitations of the study is its sample size is very small (37). It would be worth mentioning in the discussion section.

4)      Line 264 states ‘a potential for co-infection of Delta and Omicron during the time when both variants were circulating’. Please cite the reference for this.

Author Response

Reviewer 1:

 Comment #1: Line 200) states “All sample concentrations were sufficient for generating amplicon libraries for sequencing studies”. What do you mean by sample concentration? Volume? Or concentration of RNA?

Response #1: We revised in the sentence to “We report that we generated amplicon libraries from all samples with Ct values less than 33 (N1 assay). These amplicon libraries were subjected to sequencing using Illumina Miseq.” (lines 218-221).

Comment #2: In the introduction part, I would suggest adding more about the two primer schemes. Why were they chosen? What is the difference between the 2 to make it clear for the reader? What are the advantages of studying or comparing them for future studies?

Response #2: As suggested, we modified the introduction to state that we tested the two primer schemes so that we could determine if the resulting amplicons provided good breadth of coverage of both Delta and Omicron VOCs (lines 84-87).

In the discussion, we describe the advantages of testing the two primer schemes for Delta and Omicron, including that both schemes identified the same deletion in the P2 sample, and that we confirmed excellent coverage (97.5-99.9%, Figure 1) of Omicron VOCs (lines 500-505).

Comment #3: One of the biggest limitations of the study is its sample size is very small (37). It would be worth mentioning in the discussion section.

Response #3: We agree, and we addressed this limitation in the discussion (lines 517-526).

Comment #4: Line 264 states ‘a potential for co-infection of Delta and Omicron during the time when both variants were circulating’. Please cite the reference for this.

Response #4: We added the reference to the work of Rockett et al. (2022) Nature Communications (line 287), which describes detecting co-infections with Delta and Omicron in two patients with chronic kidney disease during the time when both of these variants were circulating in the local community. We also added this reference to the Results section (line 382).

Reference added:

Rockett, R.J.; Draper, J.; Gall, M.; Sim, E.M.; Arnott, A.; Agius, J.E.; Johnson-Mackinnon, J.; Fong, W.; Martinez, E.; Drew, A.P.; et al. Co-Infection with SARS-CoV-2 Omicron and Delta Variants Revealed by Genomic Surveillance. Nat. Commun. 2022, 13, 2745, doi:10.1038/s41467-022-30518-x.

Reviewer 2 Report

This study covers the sequencing detection between delta and omicron variants in clinical samples, to differentiate the two variants of concerns in a designated area. Before this study is accepted for publication, here listed several concerns:

1. Why did the study focus on breakthrough infections? It is reasonable that the breakthrough from vaccination may be caused by the mutants with hight genetic variations. To this end, control samples need to be compared and cleared out, instead of simply assumption.

2. My point of view is the sample size is relatively small. Dozens of samples can be very limited to make any meaningful conclusion. Please consider adding more samples to support your validation.

3. How do the authors explain the effect of vaccination times on mutation rate? It is known now that incomplete vaccination can not provide sufficient protection to patients, so the times of vaccination matter, in connnection to the mutation rate and extent. 

Author Response

Reviewer 2:

Comment #1: Why did the study focus on breakthrough infections? It is reasonable that the breakthrough from vaccination may be caused by the mutants with hight genetic variations. To this end, control samples need to be compared and cleared out, instead of simply assumption.

Response #1: We revised the introduction section (lines 42-45) to clarify that our focus is on breakthrough infections in immunosuppressed patients. Previous studies have shown that these patients are vulnerable to SARS-CoV-2 infections with on-going SARS-CoV-2 virus replication and are more likely to have severe disease outcomes. Therefore, studying SARS-CoV-2 variants in this population may yield new information on evolution of the entire genome of SARS-CoV-2.

We agree that a clear identification of genetic features representative for SARS-CoV-2 breakthrough infections will need the comparison from the general population. Our work aimed to report identifiable SARS-CoV-2 variants/sublineages and genetic features from immunosuppressed patients, as well as to provide information on our sequencing approaches and bioinformatics analysis for clinical samples. We report that our samples tracked Omicron community breakthrough infections in Illinois in late 2021 with the same clade-crucial mutations (C11950T and C28472T, Spencer et al. 2022; Figure 5A) (lines 442-447). The work of Spencer et al. is in line with our main finding that the variants detected in our immunosuppressed patients are consistent with the ones in local general population during the Delta and Omicron transition period (Figure 2B).

Comment #2: My point of view is the sample size is relatively small. Dozens of samples can be very limited to make any meaningful conclusion. Please consider adding more samples to support your validation.

Response #2: We agree with the reviewer that our sample set was relatively small. We have acknowledged this limitation in the discussion (lines 517-526). This work contributes to our understanding of how SARS-CoV-2 variants evolve in immunosuppressed patients.

Comment #3: How do the authors explain the effect of vaccination times on mutation rate? It is known now that incomplete vaccination can not provide sufficient protection to patients, so the times of vaccination matter, in connection to the mutation rate and extent.

Response #3: We do not address the effect of vaccination time on mutation rate in this manuscript. Our focus is to identify mutations across the entire genome of SARS-CoV-2 variants of concern after breakthrough infection in immunosuppressed patients. We detected single nucleotide variants in multiple regions across the whole genome, not only the spike gene. We revised the introduction (lines 57-59) and discussion sections (lines 463-466) to make this idea more clear.

Reference:

Spencer, H.; Teran, R.A.; Barbian, H.J.; Love, S.; Berg, R.; Black, S.R.; Ghinai, I.; Kerins, J.L.; Patrick, S.; Kauerauf, J.; et al. Multistate Outbreak of Infection with SARS-CoV-2 Omicron Variant after Event in Chicago, Illinois , USA , 2021. Emerg Infect Dis. 2022, doi:10.3201/eid2806.220411.

Reviewer 3 Report

In the study by Feng et al. entitled “Sequencing during times of change: Evaluating SARS-CoV-2 clinical samples during the transition from the Delta to Omicron wave”, the authors investigated lineages and host-specific mutations identified in a particularly vulnerable population of predominantly older and immunosuppressed SARS-CoV-2 infected patients seen at a medical center in Chicago during the transition from the Delta to Omicron wave. They compare two primer schemes, ArticV4.1 and VarSkip2, used for short read amplicon sequencing, and describe a strategy for bioinformatics analysis that facilitates identifying lineage-associated mutations and host-specific mutations that arise during infection. This study illustrates the ongoing evolution of SARS-CoV-2 VOCs in their community and documents novel constellations of mutations that arise in individual patients. The authors concluded that ongoing evaluation of the evolution of SARS-CoV-2 during this pandemic is important for informing our public health strategies. The theme is important and the manuscript should be accepted after minor revisions.

 Minor revisions:

1) Page 3, line 100: What RNA concentration was used in 5 µl of template RNA?

 2) Page 4 line 160: What RNA concentration was used in 11 µl of RNA?

 3) Despite the reduced number of samples analyzed, it is essential to understand the SARS-CoV-2 evolution in immunosuppressed patients. Furthermore, the study highlights this limitation in the discussion. Additionally, the work was well described (small changes are necessary), with adequate analysis and a clear presentation of the results. For all the reasons listed above, the work should be accepted for publication in Viruses Journal.

Author Response

Reviewer #3:

Comment #1: Page 3, line 100: What RNA concentration was used in 5 µl of template RNA?

Response #1: We revised this description to indicate that 5 µl of template RNA from all samples with the N1 gene Ct values less than 33 were used to generate the sequencing libraries.

Comment #2: Page 4 line 160: What RNA concentration was used in 11 µl of RNA?

Response #2: We revised line 174 to include the concentration of RNA.

 Comment #3: Despite the reduced number of samples analyzed, it is essential to understand the SARS-CoV-2 evolution in immunosuppressed patients. Furthermore, the study highlights this limitation in the discussion. Additionally, the work was well described (small changes are necessary), with adequate analysis and a clear presentation of the results. For all the reasons listed above, the work should be accepted for publication in Viruses Journal.

Response #3: We thank reviewer #3 sincerely for the accurate summary. Our study of SARS-CoV-2 variants in breakthrough infection of immunosuppressed individuals contributes to understanding SARS-CoV-2 viral evolution in this patient population.  

Reviewer 4 Report

1. Authors have mentioned that the swap samples were collected from patients; have authors checked their disease history and co-morbidities before collecting the samples?

2. Authors have stated that “We checked the total mutation numbers at read frequencies of 0.01, 0.03, 0.1, 0.3, 149 0.5, 0.7, 0.8, 0.95 and 1 for the eight samples sequenced”. But no information on total number sample, how do they screen the 8 samples in Read frequency cut off analysis?

3. Why authors have specifically used viral RNA isolated from P2 and P10 for the evaluation of ORF7a region. Provide this information in more detail.

4. Authors have mentioned that both delta and omicron viruses have private mutation, which one has more repeated private mutation during this analysis? Is this private mutation may influence the virulence of the virus?

5. Did authors have noticed alpha and delta variants in patients samples collected during the course time?

6. Authors have requested to provide a total number of variants that have been studied in this analysis more in detail.

7. Did authors have noticed any unique mutations in Omicron and Delta samples during the course time?

Author Response

Reviewer #4:

Comment #1: Authors have mentioned that the swap samples were collected from patients; have authors checked their disease history and co-morbidities before collecting the samples?

Response #1: The detailed meta-data for the patients is provide in Table S1, which is indicated on lines 208-212 in the Results section 3.1. Our physician co-authors reviewed the metadata carefully and identified immunosuppressed patients who were at high risk for prolonged virus replication. Patients with a recent disease history that included multiple myeloma, solid organ transplant, systemic lupus erythematosus, or chronic lymphoid leukemia, were classified as immunosuppressed in this study.

Comment #2: Authors have stated that “We checked the total mutation numbers at read frequencies of 0.01, 0.03, 0.1, 0.3, 149 0.5, 0.7, 0.8, 0.95 and 1 for the eight samples sequenced”. But no information on total number sample, how do they screen the 8 samples in Read frequency cut off analysis?

Response #2: We revised the Material and Methods section 2.5 to include this information (lines 161-163): Briefly, we called mutations for each of the 8 samples at the listed read frequencies using iVar (command “ivar variants”), and then summarize the total mutation numbers in the eight samples at each read frequency. We evaluated eight samples because these were the samples run using both Artic and VarSkip2 primer schemes. The results provided information on the mutation numbers detected in specific samples evaluated at different read depths.

Comment #3: Why authors have specifically used viral RNA isolated from P2 and P10 for the evaluation of ORF7a region. Provide this information in more detail.

Response #3: We analyzed the P2 sample because both ArticV4.1 and VarSkip2 sequencing results indicated a deletion in the ORF7a regions as shown in Figure 4. We selected the P10 sample as a control Delta sample that had no evidence of deletion by amplicon sequencing. We revised the Material and Methods section to make this clearer (lines 176-180). 

Comment #4: Authors have mentioned that both delta and omicron viruses have private mutation, which one has more repeated private mutation during this analysis? Is this private mutation may influence the virulence of the virus?

Response #4: We did not observe repeated private mutations in the Delta samples of different patients, even in those of the same sublineage (AY.3, P5, P6 and P10), indicating patient-specific mutations during SARS-CoV-2 infections. This is in agreement with the work of Lythgoe and the concept of “iSNVs” (Lythgoe et al. 2021), which we reference in the discussion (line 456). We did observe repeated private mutations in our Omicron samples, such as C21595T, C11950T, and C28472T. This repeating pattern of specific mutations in patient samples suggested that a new sublineage is starting to be transmitted in a community. We noted that the C11950T mutation we identified in our samples has since become associated to a new Omicron sublineage BA.1.20.

The question of potential impacts of these mutations on the virulence of SARS-CoV-2 is important. We note that non-spike mutations, such as those in the N gene, may contribute to enhanced transmission of VOCs (Abdullah et al. 2021; discussion section lines 463-466). Continuous genomic surveillance to identify mutations in clinical samples along with patient health status will help us understand the impact of the mutations on transmission and virulence of emerging VOCs.

Comment #5: Did authors have noticed alpha and delta variants in patients samples collected during the course time?

Response #5: From late 2021 to early 2022, we collected Delta and Omicron variants infected samples in our medical center, but no Alpha, corresponding to the variants circulating in the local communities (Figure 2B). We did collect one Alpha in May 2021 (sample P1); this sample was sequenced successfully and served as a quality control for the sequencing and bioinformatic analysis in this study.

Comment #6: Authors have requested to provide a total number of variants that have been studied in this analysis more in detail.

Response #6: We have provided the detailed list of variants, sublineages and private mutations in Table S2.

Comment #7: Did authors have noticed any unique mutations in Omicron and Delta samples during the course time?

Response #7: Yes. We described the private mutations found in our Delta and Omicron samples in Figure 5A and Table S2. These private mutations can be viewed as “sample-unique” mutations that are not prevalent in the Delta or Omicron variants reported globally, supporting the finding of “iSNVs” (Lythgoe et al. 2021).

Reference:

Lythgoe, K.A.; Hall, M.; Ferretti, L.; de Cesare, M.; MacIntyre-Cockett, G.; Trebes, A.; Andersson, M.; Otecko, N.; Wise, E.L.; Moore, N.; et al. SARS-CoV-2 within-Host Diversity and Transmission. Science 2021, 372, eabg0821, doi:10.1126/SCIENCE.ABG0821.

Reference added:

Abdullah, M.S.; Taha, T.Y.; Tabata, T.; Chen, I.P.; Ciling, A.; Khalid, M.M.; Sreekumar, B.; Chen, P.; Hayashi, J.M.; Soczek, K.M.; et al. Rapid Assessment of SARS-CoV-2–Evolved Variants Using Virus-like Particles. Science 2021, 374, 1626–1632, doi:10.1126/SCIENCE.ABL6184.

Round 2

Reviewer 4 Report

Revised version is satisfactory, and in present form, the manuscript can be accepted for publication. 

Best wishes for the authors